# Health Literacy as a Shared Capacity: Does the Health Literacy of a Country Influence the Health Disparities among Immigrants?

**DOI:** 10.3390/ijerph17041149

**Published:** 2020-02-12

**Authors:** Chiara Lorini, Saverio Caini, Francesca Ierardi, Letizia Bachini, Fabrizio Gemmi, Guglielmo Bonaccorsi

**Affiliations:** 1Department of Health Science, University of Florence, 50134 Florence, Italy; guglielmo.bonaccorsi@unifi.it; 2Institute for Cancer Research, Prevention and Clinical Network (ISPRO), 50139 Florence, Italy; s.caini@ispro.toscana.it; 3Quality and Equity Unit, Regional Health Agency of Tuscany, 50141 Florence, Italy; francesca.ierardi@ars.toscana.it (F.I.); letizia.bachini@ars.toscana.it (L.B.); fabrizio.gemmi@ars.toscana.it (F.G.)

**Keywords:** health inequalities, health literacy, immigrant, population

## Abstract

Health literacy (HL) is an individual ability as well as a distributed resource available within an individual’s social network. We performed an explorative study assessing the role of HL as the country-level ecological variable in predicting the health disparities among immigrants. Country-level HL data were obtained from the publicly available first European Health Literacy Survey reports. Individual-level data on citizenship, perceived health status, body mass index, smoking habits, physical activity and attendance at breast and cervical cancer screening were extracted from the European Health Interview Survey of Eurostat. Data from both sources were obtained for Austria, Bulgaria, Greece, Poland and Spain. The country-specific odds ratio (OR) for the association between the participants’ citizenship and other individual health-relevant characteristics was pooled into summary OR using random-effects models. Meta-regression was used to explore whether the HL of a country could explain part of the between-countries heterogeneity. Results: For the perceived health status, nutritional status and attendance at cervical cancer screening, the lower was the country-level HL (as ecological variable), the higher were the health inequalities relating to citizenship. The results of our exploratory research suggest that improving the population HL may help mitigate health inequalities between residents and migrants.

## 1. Introduction

Health literacy (HL) is gaining critical importance in public health. From the original perspective focusing on handling words and numbers in the medical context, the concept has gradually expanded to include more complex abilities relating to information-seeking, decision-making, problem-solving, critical thinking and communication, all of which are crucial in interacting with the health system [1]. Freedman [2] defined public heath literacy as “the degree to which individuals and groups can obtain, process, understand, evaluate, and act information needed to make public health decision that benefit the community”. This is distinct from, yet related to, individual-level HL. This concept is strongly linked with that of distributed HL [3]: according to Papen [4], HL is not only an individual ability, but also distributed resource available within an individual’s social network. In fact, health decisions are often not made by individuals alone but collectively—they involve other people in the decision-making process and there are cultural differences in how decisions are made [5]. Batterham [6] stressed the importance of a distributed HL both for individual empowerment (freedom of choices and participation in decision-making) and adherence to professional medical advice. Thus, the HL abilities, skills and experiences of other people can play a role in improving individual HL and compensate for an individual’s poor HL [3], affecting both pathways’ individual health-related behaviours and health outcomes, as suggested also by some published research studies on patients [7,8], communities [9] and populations [10]. Specifically, in our previous ecological study conducted at the national level [10], the mean HL was found to be correlated with national indicators related to consumer empowerment, the prevalence of overweight individuals, the health status of the population and the total health expenditure. Moreover, in the same study, the mean HL was correlated with the indicators related to the prerequisites of health (specifically education, income, social justice and equity) and the health coverage (the presence of national screening programmes), thereby empowering the healthcare systems for patients. In this sense, the HL of a country could be considered as the effect of many national policies devoted to improving population health and reducing inequalities.

People with a restricted social network and a low level of integration within the community may be the most disadvantaged in terms of developing individual HL or joining support for others’ HL. This may contribute to health inequalities. For immigrants, these aspects are particularly relevant due to linguistic and cultural barriers [11,12]. Moreover, since HL is the balance of individual skills and the demands coming from the context, it may change with time and circumstances, thus people may be considered health-literate in one country but not in another [5].

Many studies have investigated the health inequalities of immigrants, highlighting that they are present worldwide in varying levels across contexts [13,14,15,16]. Other researchers have investigated the HL in such target groups as well as the relationship between the individual HL of immigrants and health outcomes [17,18,19]. To the best of our knowledge, no studies have been published addressing the health-related behaviours and outcome of immigrants with respect to the HL context. This study aimed to assess the role of HL as the country-level ecological variable in predicting the health disparities among immigrants in different EU countries. The research question was: Does the HL of a country influence the health disparities among immigrants? This exploratory research provides a novel approach to study this phenomenon to advance our understanding of the role of distributed HL.

## 2. Materials and Methods

The study was performed considering the HL as a country-level ecological variable and keeping the other variables at the individual level. Data were collected using pre-existing sources.

### 2.1. Data Source for HL

Information on HL was obtained from the published results of the first European HL Survey (HLS-EU) [20,21,22]. That survey was conducted in 2011 as a population study according to Eurobarometer standards in eight European countries (Austria, Bulgaria, Germany, Greece, Ireland, the Netherlands, Poland and Spain). Randomly selected sampling points were used from each administrative region in a country. Two exemptions were made for logistical and cost-efficiency reasons. Germany was only represented by its most populated federal state, North-Rhine Westphalia. In Greece, following general Eurobarometer practice, the survey collected data in greater Athens [20]. HL was measured using the HLS-EU-Q47 tool, which is based on the conceptual model developed by Sørensen et al. [23,24]. The HLS-EU-Q47 consists of 47 items, to which the interviewees responded by rating the perceived difficulty of a given task [23,24]. Answers to each item are on a four-category Likert scale (from “very easy” to “very difficult”), while the score from the 47 items are converted into a total HL score (whose values range from 0 to 50) that is used to categorise the respondents into different levels of HL (“inadequate”, “problematic”, “sufficient” and “excellent”).

### 2.2. Data Source for Sociodemographic, Health-Related Behaviours and Outcome Variables

Individual data were extracted from the database of the European Health Interview Survey (EHIS) of Eurostat. (Eurostat provides access to European microdata for research purposes. The access is provided to allow scientific research for the benefit of society. That is why Eurostat asks researchers to send the publications produced on the basis of the microdata. This generates a list of research publications. The list gives an indication of the impact of the Eurostat microdata access system. It helps researchers to formulate new and innovative scientific questions. Researchers must ensure that any results of the research published or otherwise disseminated do not contain information that may permit the identification of individual statistical units (persons, households, enterprises, etc.) (https://ec.europa.eu/eurostat/web/microdata/overview).) The EHIS is a general population survey that provides comparable cross-national data on the health status, healthcare and health determinants in the European Union (EU). Since the HLS-EU survey was conducted in 2011, EHIS data of the first wave (2006–2009) were requested and included in the analysis. Since two of the eight countries involved in the HLS-EU survey (Ireland and The Netherland) did not join the first wave of the EHIS and another one (Germany) did not grant access to the microdata collected in the survey, we only included in this study the following five countries: Austria, Bulgaria, Greece, Poland and Spain.

EHIS data included in this study regard sociodemographic, health-related characteristics and behavioural information. In Table 1, all the variables extracted from EHIS are described.

### 2.3. Statistical Analysis

The distribution of the main participants’ characteristics in tables, using the chi-square tests to compare their distribution across countries, is shown.

The statistical analysis was conducted in two successive steps. First, we fitted country-specific logistic regression models to study the association between citizenship (local citizens, migrants from EU and non-EU countries) and the following health-related characteristics and behaviours: perceived health status (very good or good vs. fair, bad, or very bad), body mass index (≥25 vs. <25 kg/m^2^), smoking habits (smoker vs. non-smoker), physical activity (physically active vs. inactive) and attendance at breast and cervical cancer screening (yes vs. no). All country-specific models were adjusted by gender, age, marital status, educational level (highest degree attained), employment status, urbanisation of the area of residence and presence of chronic illnesses and medical conditions. Owing to the low percentage of missing values for most variables of interest (see Section 3), we conducted a complete case analysis—we made no attempt to impute missing values.

Odds ratios (OR) and the corresponding variance from country-specific logistic models were then pooled into summary odds ratio (SOR) and the corresponding 95% CI using random effect models with maximum likelihood estimation and assuming an underlying t distribution [25]. We assessed the between-studies heterogeneity using the I2 statistics, a measure of the proportion of the variation of the effect across countries which is attributable to actual heterogeneity rather than chance [26]. Larger values of the I2 statistics denote greater between-countries heterogeneity; a value of I2 below 50% is considered as an acceptable degree of heterogeneity. When the I2 statistics exceeded 50%, we visually inspected the forest plots to evaluate whether the large heterogeneity could be caused by a single country having an OR very different from those of the other countries and, accordingly, used meta-regression to explore whether the country-level HL (measured ecologically using the HLS-EU-Q47: average value and proportion of the country population with values of health literacy judged as “inadequate” or “problematic or inadequate”) could explain part of the between-studies heterogeneity.

Statistical analyses were conducted using the SAS software, version 9.2 (SAS Institute Inc, Cary, NC, USA). All tests were two-sided and considered as statistically significant for *p*-values lower than 0.05.

## 3. Results

The study participants differed substantially across countries (*p*-value < 0.001) in terms of characteristics that were used in subsequent analysis, including the demographics age group and gender, citizenship, educational level and employment status, marital status, degree of urbanisation of their area of residence, health status (underlying illness or health problem and perceived health status), anthropometry, lifestyle (smoking habits and physical activity levels) and, among women, attendance at cancer screening programmes (breast and cervical cancer) (Table 2). The proportion of missing values was below 1% for age, gender, citizenship, educational level, employment status, marital status, degree of urbanisation and the presence of underlying illnesses or health problems; between 1% and 5% for physical activity levels and attendance at breast or cervical cancer screening; and above 5% for perceived health status (5.5%), body mass index (9.2%) and smoking habits (6.1%).

HL (as measured by the HLS-EU-Q47) differed significantly across countries, both when considering the proportion of a country population falling in each of the four classes (*p* < 0.001) and when considering its average value (*p* < 0.0001) (Table 3). In particular, the average HLS-EU-Q47 value and the proportion of population with excellent or sufficient HL were highest in Poland (34.4% and 55.4%, respectively) and lowest in Bulgaria (30.5 and 37.9%, respectively) (Table 3).

### 3.1. Perceived Health Status

Compared to local citizens, the likelihood of reporting very good or good (vs. fair, bad or very bad) health status was significantly lower among nationals of other EU countries (SOR 0.75, 95% CI 0.61–0.92) and, at an even higher degree, among citizens of non-EU countries (SOR 0.54, 95% CI 0.39–0.75) (Table 4 and Figure A1).

In the latter case, the heterogeneity among countries was substantial (I2 63.4%): the visual inspection of the forest plot suggested that Greece behaved as an outlier, as its OR was closer to the null value (and not significantly different from it: OR 0.77, 95% CI 0.48–1.22) compared to the other countries. In meta-regression models, the difference in the perceived health status between local citizens and migrants from non-EU countries was milder (with borderline statistical significance: *p*-value 0.056) in countries where the average HL (as measured using the HLS-EU-Q47) was higher and vice versa (Table 5).

### 3.2. Body Mass Index

The likelihood of being overweight or obese (vs. normal-weight) did not significantly deviate from the null value among foreign citizens of both the EU and non-EU states (compared to local citizens): the SOR was 0.79 (95% CI 0.60–1.06) for the former and 1.20 (95% CI 0.80–1.81) for the latter (Table 4 and Figure A2). However, the heterogeneity of ORs across countries was above the threshold of acceptability in both cases (I2 equalled 55.2% and 87.42%, respectively). Among foreign EU citizens, the country-specific OR fluctuated in a wide range, between a minimum of 0.25 (95% CI 0.07–0.96) in Bulgaria and a maximum of 2.16 (95% CI 0.63–7.41) in Poland. The OR for nationals of non-EU countries ranged between 1.39 and 1.71 for all countries (achieving statistical significance in Austria and Spain), except Greece, where, on the contrary, the OR was significantly smaller than unity (0.65, 95% CI 0.49–0.85). In meta-regression models, the adjusted country-specific OR of being overweight/obese among migrants from non-EU countries (compared to local citizens) tended to be positively correlated, although not achieving statistical significance (*p*-value 0.075), with the proportion of the population showing their HL considered as “inadequate” or “problematic” (Table 5). No such correlation emerged in case of any comparison between local citizens and migrants from other EU countries.

### 3.3. Smoking Habits

Migrants from other EU countries were less likely to be smokers than local citizens (SOR 0.65, 95% CI 0.50–0.85) (Table 4). The heterogeneity between countries was slightly below the threshold of acceptability (I2 = 46.9%). However, the visual inspection of the forest plot (Figure A3) revealed that country-specific OR was significantly lower than the null value in Austria, Greece and Spain (0.68, 0.51 and 0.59, respectively), while it was above 2.0 (although not achieving statistical significance) in Bulgaria (2.08, 95% CI 0.36–11.98) and Poland (2.90, 95% CI 0.69–12.23). The likelihood of being a smoker did not differ among citizens of non-EU countries (compared to local citizens) in the pooled estimate (SOR 1.10, 95% CI 0.57–2.11). However, country-specific ORs differed widely among one another (from a minimum of 0.56 in Poland to a maximum of 1.96 in Greece) with very large between-countries heterogeneity (I2 = 95.5%). A further element of variability could be seen in the fact that the OR of being a smoker for citizens of the EU and non-EU countries were both above the null value (i.e., >1.00) in Bulgaria; diverged in Greece, Poland and Spain; and below the null value (i.e., <1.00) in Austria. The diversity of countries in terms of HL was unable to account for the heterogeneity of the country-specific OR to be a smoker in any of the meta-regression models that we fitted (Table 5).

### 3.4. Physical Activity Levels

The likelihood of being physically active did not significantly differ between local citizens and migrants from another EU country (SOR 1.10, 95% CI 0.81–1.50) or from non-EU countries (SOR 0.95, 95% CI 0.56–1.62) (Table 4). In the latter case, however, the country-specific OR differed widely between one another (I2 = 95.5%), as migrants were ≈ 50% less likely to be physically active than local citizens in Austria and Bulgaria (significantly so in the former country) and more than twice as likely in Greece (achieving statistical significance: OR 2.13, 95% CI 1.40–3.25) (Figure A4). This disparity was partially accounted for by diversity in the HL across countries (Table 4). In detail, the odds of a migrant to be sedentary tended to be greater in countries with a lower average HLS-EU-Q47 score (*p*-value 0.065) and the physical activity of migrants was a more common finding in countries with higher average HLS-EU-Q47 score.

### 3.5. Attendance at Breast Cancer Screening Programmes

Participation in organised mammography screening was less frequent among migrants from other EU countries compared to local citizens (SOR 0.66, 95% CI 0.31–1.39); it is even lower (and achieving statistical significance) among migrants from non-EU countries (SR 0.34, 95% CI 0.16–0.71) (Table 4). Heterogeneity between countries was high (I2 > 60%). However, for both comparisons, an OR above 1.00 (i.e., denoting a higher participation in the breast cancer screening programme compared to local citizens) emerged among EU migrants living in Bulgaria (2.52, 95% CI 0.40–15.76) and Greece (1.19, 95% CI 0.23–6.10), as well as among non-EU migrants living in Poland (1.26, 95% CI 0.12–13.69) (Figure A5). The country-specific OR was not associated with the country-level HL in none of the meta-regression models that were fitted (Table 5).

### 3.6. Attendance at Cervical Cancer Screening Programmes

The likelihood of participating in cervical cancer screening programmes was significantly lower among migrants from both other EU (SOR 0.61, 95% CI 0.49–0.77) and non-EU countries (SOR 0.38, 95% CI 0.20–0.73) (Table 4). The heterogeneity between countries was large (I2 = 89.9%) only in the latter case, but the visual inspection of the forest plot suggested that this was mainly the consequences of very narrow 95% CI in country-specific estimates as the OR was below the null value and ranged between 0.18 (in Austria) and 0.63 (in Spain) (Figure A6). In meta-regression models, the higher the proportion of a country population with their HL judged as “problematic”, the lower the odds that non-EU migrants participated in cervical cancer screening programs (compared to local citizens) (Table 5).

## 4. Discussion

This study aimed to assess the role of HL as the ecological variable, which means that HL is considered as the macro-level aspect in predicting health disparities among immigrants at the national level. Our intention was to perform an explorative study aiming in assessing the impact of the HL of a country on health inequalities of immigrants. To the best of our knowledge, no previous studies using this study design and methodological approach have been published yet, thus comparisons with similar researches are hindered.

The results of this study show inequalities in health with regard to citizenships in five European countries (Austria, Bulgaria, Greece, Poland and Spain). For many indicators, migrants from other EU countries present worse health characteristics than nationals, and such disparities are even stronger among migrants from non-EU countries. Specifically, migrants have a worse perceived health status, a higher prevalence of overweight or obesity and a reduced attendance at breast and cervical cancer screening programmes. For smoking habits and physical activity levels, the high variability between countries limit the possibility to highlight statistically significant trends. For some of the indicators regarding health, health behaviours and use of health services applied in this study (perceived health status, nutritional status and attendance at cervical cancer screening tests), health inequalities relating to citizenship tend to be higher when the HL of the population (as ecological variable) happens to be lower.

Many studies have described health inequalities in Europe in terms of citizenship relating to health indicators and behaviours as well as to preventive programme attendance [13,27]. However, differences vary according to the specific group being studied, the health problems or services involved and the country concerned. Migrants tend to occupy a less favourable social position because of their socioeconomic status and the social exclusion they often are exposed to, with consequences on their health status [14]. Therefore, migration is increasingly recognised as an independent social determinant of health [14]. Moreover, migrant integration policies influence migrant health disparities: Policies aiming at improving opportunities for full social participation are proved to be a key factor for good health [15,28,29].

HL is a complex concept. Although most of the published papers focus on the individual’s ability, some authors have emphasised the concept of community and population HL, albeit in the absence of shared definitions and any specific measuring instrument [30]. At the population level, many determinants could affect HL. In a previous study of our research group conducted in 2018 considering HL as a country-level ecological variable, it emerged that national policies devoted to promoting and providing the prerequisites of health (education, income, social justice and equity), increasing the health coverage (the introduction of national screening programs) and making healthcare systems more empowering for patients result in a widespread increase in HL among the population of a nation [10]. Considering the results of this study, it should be assumed that the same determinants of population HL could contribute in reducing health inequalities, including those related to citizenship. Moreover, it should be noted that Austria and Bulgaria, which presented the highest percentage of people with inadequate HL, also had more problematic integration policies [28].

The results of this study are in line with the conceptualisation of public HL. According to Freedman [2], this conceptualisation includes the basic knowledge and information needed to understand and take action on public health concerns. Three dimensions of public HL are identified: conceptual foundations, critical skills and civic orientation. Health-literate populations have critical skills necessary to obtain, process, evaluate and act upon information needed to make public health decisions that would benefit the community and present a civic orientation ensuring that the public interest remains at the centre. Moreover, health-literate populations have the skills and resources necessary to address health concerns through civic engagement. Public HL is as much citizen-based as it is expert-driven and it can take multiple forms, from voting to organising grass-roots initiatives to establishing healthy policies and structures. The target population for promoting public HL is the entire public, not just public health and medical officials. An individual or group demonstrating public HL from a civic perspective is able to: articulate that the burdens and benefits of society are not fairly distributed; evaluate who benefits and who is harmed by public health efforts or lack thereof; communicate that current public health problems are not inevitable and can be changed through civic action; and address public health problems through civic action, leadership and dialogue. Civic engagement calls for awareness of the ways in which public goods, resources, burdens and benefits are distributed, and it is the first step towards civic action to advocate on behalf of public health [2]. Thus, health-literate populations are more oriented towards human rights and social justice issues. In this sense, it can be assumed health-literate populations present a civic orientation more devoted to integrating migrants and reducing health inequalities. Critical health literacy, in particular, enhances the capacity to act politically to address social and economic determinants of health; incorporates the knowledge of social processes, social capital, social cohesion and media literacy skills; and contributes to community empowerment [23,31,32,33]. In this regard, both community and population HL may be viewed as an asset to overcome most of the barriers to access and participation by using appropriate communication strategies, working with cultural factors that influence behavioural change and assessing affiliation with cultural norms. In fact, many studies have suggested that migrants benefit from better information on health services and entitlements, as well as from education programmes to improve HL [34].

The limits of this study are partly related to the limited number of countries involved. This is the consequence of the restricted number of countries included in the first European HL survey (*n* = 8) and the exclusion of three of them due to unavailability of data in the EHIS database. The foreseen inclusion of many other countries in the next European HL survey [35] that will be conducted shortly, will result in the availability of HL data for many more European countries, which can be therefore included in future research conducted with the same methodological approach. Moreover, EHIS data referred to adults, especially regularly residents in European countries, thus migrants with more social exclusion—those who are not residents of those countries—are not included in this study. Consequently, health inequalities relating to citizenship are presumably underestimated in this research. Additionally, the EHIS does not include a measure of HL, thus individual data on this variable were not available. Furthermore, in the HLS-EU survey, the sampling procedure presented some differences in the five countries included in this study, limiting the generalisability of the results of the HL data for some countries. Besides, HL was considered as an ecological variable, measured either as the mean score at the HLS-EU-Q47, the percentage of population with inadequate HL, or the percentage of those with problematic or inadequate HL. The ecological approach allowed us to examine how the shared HL of the population affects health inequalities relating to citizenship, with the limit of making inferences at the individual level. On the other hand, the latter was not part of the aim of this study, since our purpose was to assess HL as a shared capacity and not as an individual skill.

Other limits are related with not having included in the model other country-level variables that could affect both inequalities in immigrants’ health and the national level of HL: the purpose of this research was also to test a new, different approach in analysing such a multifaceted issue, using a synthetic, although a bit reductive, model. Future studies will be performed to deeply investigate the network of determinants of health inequalities for immigrants, including HL at both the country and the individual level, as well as other country-level variables.

## 5. Conclusions

Despite the limited number of countries included in this exploratory study, the limitation in the study design and the non-inclusion of non-resident migrants, it should be noted that the HL of the population may affect health inequalities relating to citizenship. Therefore, improving the population HL may be worth exploring to help reduce health inequalities of migrants. Both population and community HL should be important future research areas to resolve the persistent ethnic health disparities. More discussion and research are needed to improve our understanding of the role of HL in the context. Future research should aim to find out how to reduce the situational demands and complexity in which an individual makes a health decision, including how organisations and social institutions can contribute in this regard.

## Figures and Tables

**Table 1 ijerph-17-01149-t001:** Sociodemographic, health-related behaviours and outcome variables from the European Health Interview Survey (EHIS) of Eurostat, included in the study (individual level).

Variables	Description in the European Health Interview Survey (EHIS) Database (Questions to Participants and Answers) and Categorisation for This Study
Age	Age of the person at the moment of interview Modes: the numerical value has been ranged into four classes (15–29; 30–44; 45–60; 60+)
Gender	Modes: “Male”, “Female”
Citizenship	What is your citizenship? Modes: “nationals”, “nationals of other EU Member States”, “nationals of non-EU countries”
Degree of urbanisation of the area of residence	Three modes based on a combination of geographical contiguity and population density, measured by minimum population thresholds applied to 1 km^2^ population grid cells: “densely-populated area”, “Intermediate area”, “thinly-populated area”
Legal marital status	What is your legal marital status? Modes: “never married”, “married”, “widows and not remarried” and “divorced and not remarried”
Education level	What is the highest education leaving certificate, diploma or education degree you have obtained? Seven modes summarised into three final modes following the International Standard Classification of Education: “Less than primary, primary and lower secondary education”, “Upper secondary and post-secondary non-tertiary education”, “Tertiary education”
Employment status	How would you define your current labour status? Eight modes summarised into three final modes: “worker”, “not a worker”, “former worker”
Presence of chronic illnesses	Do you have any longstanding illness or (longstanding) health problem? Modes: “yes”, “no”
Body Mass Index (BMI)	How much are you weight without shoes and shoes? and How tall are you without shoes? BMI calculated as weight/height^2^, and then categorised into two categories: underweight/normal subject” (BMI ≤ 25 kg/m^2^) and “overweight/obese subject” (BMI ≥ 25 kg/m^2^)
Physical activity	During the last 7 days, on how many days did you do vigorous physical activities? During the last 7 days, on how many days did you do moderate physical activities? The answers to the two questions were combined in a dichotomous variable, with “Active” and “Inactive” modes. The “Active” mode included those who stated that they performed moderate physical activity at least 2/3 times a week or intense at least 1/2 times a week. The “Inactive” mode included those who stated that they performed moderate physical activity less than 2/3 times a week or intense less than 1/2 times a week
Smoking habits	Do you smoke at all nowadays? Modes: “regularly”, “occasionally”, “never”. Modes dicotomised as “Smoker” (both regular and occasional) and “Non-smoker”
Adherence to breast cancer screening	Have you ever had a mammography, which is an X-ray of one or both of your breasts? Modes: “yes”, “no” The women ≥50 years were selected. The women who declared they had adhered to the screening every two years have been included in the “Adherent” mode; the women who have declared that they have not performed screening or have performed it at intervals of more than two years have been included in “Non-adherent” mode.
Adherence to cervical smear screening	Have you ever had a cervical smear test? Modes: “yes”, “no” The women ≥ 25 years were selected. The women who declared they had adhered to the screening every two years have been included in the “Adherent” mode; the women who have declared that they have not performed screening or have performed it at intervals of more than two years have been included in “Non-adherent” mode.
Perceived Health Status	How is your health in general? Five modes: “very good”, “good”, “fair”, “bad”, and “very bad”. The final variable was dichotomous: “good” (very good, good) and “bad” (fair, bad, very bad).

**Table 2 ijerph-17-01149-t002:** Characteristics of study participants, overall and by country. Data from European Health Interview Survey (EHIS), 2006–2009, Eurostat.

Variables		Total ^(1)^ (*n* = 84,503)	Austria ^(1)^ (*n* = 15,382)	Bulgaria ^(1)^ (*n* = 5661)	Greece ^(1)^ (*n* = 6172)	Poland ^(1)^ (*n* = 35,100)	Spain ^(1)^ (*n* = 22,188)	*p*-Value ^(2)^
**Age (years)**	15–29	26.6%	27.1%	23.1%	20.7%	31.4%	21.2%	<0.001
	30–44	16.7%	18.9%	14.0%	16.0%	14.5%	19.7%	
	45–60	33.3%	31.2%	36.2%	30.4%	35.2%	32.0%	
	60+	23.3%	22.8%	26.7%	32.9%	18.9%	27.2%	
**Gender**	Male	45.3%	45.4%	47.4%	39.1%	46.0%	45.3%	<0.001
	Female	54.7%	54.6%	52.6%	60.9%	54.0%	54.7%	
**Citizenship**	Nationals	96.9%	94.5%	99.6%	94.5%	99.9%	93.8%	<0.001
	Nationals of other EU countries	1.0%	1.8%	0.2%	1.3%	0.1%	2.2%	
	Nationals of non-EU countries	2.1%	3.7%	0.2%	4.2%	0.1%	4.0%	
**Education level** **(highest attained)**	Less than secondary education	23.4%	0.9%	8.9%	37.9%	21.1%	42.3%	<0.001
	Upper secondary and post-secondary non-tertiary	55.5%	71.6%	72.5%	38.2%	61.5%	35.5%	
	Tertiary	21.1%	27.5%	18.5%	23.7%	17.5%	22.2%	
**Employment status**	Worker	46.3%	52.4%	47.9%	38.4%	45.6%	45.0%	<0.001
	Not a worker	22.8%	18.1%	17.2%	28.7%	21.6%	27.7%	
	Former worker	30.9%	30.0%	35.9%	33.7%	32.0%	27.8%	
**Marital status**	Never married	25.4%	28.4%	21.7%	20.7%	24.7%	26.5%	<0.001
	Married	57.9%	56.6%	58.4%	60.1%	60.3%	54.1%	
	Widowed, not remarried	11.7%	8.9%	14.7%	15.5%	10.8%	13.0%	
	Divorced, not remarried	5.1%	6.1%	5.2%	3.7%	4.1%	6.3%	
**Degree of urbanisation of the area of residence**	Densely-populated area	36.8%	22.4%	43.9%	42.5%	32.9%	49.5%	<0.001
	Intermediate area	16.5%	23.7%	7.8%	11.6%	12.9%	20.6%	
	Thinly-populated area	46.8%	54.0%	48.2%	45.9%	54.2%	29.9%	
**Underlying illness or health problem**	Yes	49.6%	39.0%	43.2%	48.2%	51.1%	56.5%	<0.001
	No	50.4%	61.2%	56.6%	51.9%	48.7%	43.7%	
**Perceived health status**	Good/very good	35.1%	26.8%	36.5%	34.8%	40.5%	32.2%	<0.001
	Fair/bad/very bad	64.9%	79.0%	60.7%	71.0%	52.7%	73.6%	
**Body mass index**	Underweight/normal	37.2%	45.8%	36.1%	34.6%	34.4%	36.5%	<0.001
	Overweight/obesity	62.8%	64.4%	61.6%	71.3%	59.1%	65.7%	
**Smoking habits**	Smoker	27.8%	23.4%	37.8%	36.2%	26.2%	28.6%	<0.001
	Non-smoker	72.2%	83.1%	68.2%	70.3%	67.6%	73.5%	
**Physical activity**	Active	34.7%	26.7%	36.9%	28.7%	30.1%	48.7%	<0.001
	Inactive	65.3%	75.8%	53.4%	73.5%	68.4%	53.7%	
**Attendance of breast cancer screening**	Adherent	45.5%	68.8%	15.8%	31.9%	42.6%	48.0%	<0.001
	Non-adherent	54.5%	35.1%	81.8%	71.2%	53.4%	54.8%	
**Attendance of cervical cancer screening**	Adherent	52.8%	78.8%	28.7%	47.2%	51.8%	44.7%	<0.001
	Non-adherent	47.2%	26.2%	65.1%	57.3%	43.7%	58.5%	

^(1)^ Percentages were calculated over all participants with non-missing values. ^(2)^
*p*-Values were from chi-square test comparing the distribution of each variable across countries.

**Table 3 ijerph-17-01149-t003:** Health literacy (HL) of the general population measured during the first European HL survey (HLS-EU) in the five European countries included in the study. Mean values and breakdown into decreasing HL levels. Data from HLS-EU Consortium [22].

Variables	Austria	Bulgaria	Greece	Poland	Spain
No. participants	978	925	998	921	974
*Health Literacy (HLS-EU-Q47)*					
Excellent HL (%)	9.9%	11.3%	15.6%	19.5%	9.1%
Sufficient HL (%)	33.7%	26.6%	39.6%	35.9%	32.6%
Problematic HL (%)	38.2%	35.2%	30.9%	34.4%	50.8%
Inadequate HL (%)	18.2%	26.9%	13.9%	10.2%	7.5%
Mean value	31.9	30.5	33.6	34.4	32.9

**Table 4 ijerph-17-01149-t004:** Country-specific and summary odds ratio for the association between citizenship and the following health-related characteristics and behaviours: perceived health status, body mass index, smoking habits, physical activity and attendance of breast and cervical cancer screening.

Citizenship	Austria		Bulgaria		Greece		Poland		Spain		Pooled Estimate		
	OR ^(1)^	95% CI	OR ^(1)^	95% CI	OR ^(1)^	95% CI	OR ^(1)^	95% CI	OR ^(1)^	95% CI	SOR ^(2)^	95% CI	I^2^
Perceived health status (very good or good vs. fair, bad, or very bad)													
nationals	1.00		1.00		1.00		1.00		1.00		1.00		
nationals of other EU countries	0.81	0.57–1.16	0.74	0.22–2.46	0.59	0.32–1.08	0.82	0.27–2.44	0.74	0.55–1.00	0.75	0.61–0.92	0.0%
nationals of non-EU countries	0.39	0.31–0.49	0.43	0.06–3.10	0.77	0.48–1.22	0.56	0.16–1.90	0.61	0.49–0.76	0.54	0.39–0.75	63.4%
Body mass index (≥25 vs. <25 kg/m^2^)													
nationals	1.00		1.00		1.00		1.00		1.00		1.00		
nationals of other EU countries	0.78	0.60–1.01	0.25	0.07–0.96	0.61	0.39–0.95	2.16	0.63–7.41	0.94	0.70–1.14	0.79	0.60–1.06	55.2%
nationals of non-EU countries	1.65	1.36–2.01	1.71	0.36–8.11	0.65	0.49–0.85	1.37	0.53–3.54	1.39	1.18–1.63	1.20	0.80–1.81	87.4%
Smoking habits (smoker vs. non-smoker)													
nationals	1.00		1.00		1.00		1.00		1.00		1.00		
nationals of other EU countries	0.68	0.52–0.90	2.08	0.36–11.98	0.51	0.32–0.82	2.90	0.69–12.23	0.59	0.48–0.71	0.65	0.50–0.85	46.9%
nationals of non-EU countries	0.58	0.48–0.70	1.40	0.27–7.26	1.96	1.48–2.60	0.56	0.24–1.31	1.73	1.46–2.04	1.10	0.57–2.11	95.5%
Physical activity (active vs. inactive)													
nationals	1.00		1.00		1.00		1.00		1.00		1.00		
nationals of other EU countries	0.80	0.60–1.07	0.89	0.23–3.51	1.39	0.72–2.66	2.02	0.51–7.93	1.27	1.05–1.55	1.10	0.81–1.50	49.8%
nationals of non-EU countries	0.49	0.41–0.60	0.50	0.15–1.61	2.13	1.40–3.25	1.47	0.57–3.81	0.94	0.82–1.09	0.95	0.56–1.62	92.5%
Attendance of breast cancer screening (yes vs. no)													
nationals	1.00		1.00		1.00		1.00		1.00		1.00		
nationals of other EU countries	0.82	0.46–1.45	2.52	0.40–15.76	1.19	0.23–6.10	0.38	0.05–3.07	0.30	0.17–0.52	0.66	0.31–1.39	60.6%
nationals of non-EU countries	0.55	0.30–1.02	-	-	0.28	0.11–0.72	1.26	0.12–13.69	0.17	0.09–0.32	0.34	0.16–0.71	64.0%
Attendance of cervical cancer screening (yes vs. no)													
nationals	1.00		1.00		1.00		1.00		1.00		1.00		
nationals of other EU countries	0.66	0.42–1.01	1.25	0.26–5.97	0.81	0.38–1.72	0.94	0.18–4.91	0.55	0.41–0.74	0.61	0.49–0.77	0.0%
nationals of non-EU countries	0.18	0.13–0.25	0.32	0.05–2.06	0.41	0.28–0.62	0.55	0.15–2.06	0.63	0.51–0.78	0.38	0.20–0.73	89.9%

^(1)^ Country-specific odds ratio (OR) and corresponding 95% confidence intervals (CI) from multivariable logistic regression models further adjusted for gender, age, marital status, educational level, employment status, urbanisation of the area of residence and presence of chronic illnesses. See text for details. ^(2)^ Summary odds ratio (SOR) and corresponding 95% CI were calculated using random effect meta-analysis models.

**Table 5 ijerph-17-01149-t005:** Meta-regression models aiming at evaluating whether the country health literacy (ecological values, measured using the HLS-EU-Q47) explained the variability between countries of the association between citizenship and the following health-related characteristics and behaviours: perceived health status, body mass index, smoking habits, physical activity and attendance of breast and cervical cancer screening.

Citizenship	SOR	95% CI	I^2^	Explanatory Variable in Meta-Regression Models ^(1)^		
				Health Literacy (HLS-EU-Q47)		
				Continuous	% Population with Inadequate HL ^(2)^	% Population with Problematic or Inadequate HL ^(2)^
Perceived health status (very good or good vs. fair, bad, or very bad)						
nationals	1.00					
nationals of other EU countries	0.75	0.61–0.92	0.0%	-	-	-
nationals of non-EU countries	0.54	0.39–0.75	63.4%	0.056	ns ^(3)^	ns ^(3)^
Body mass index (≥25 vs. <25 kg/m^2^)						
nationals	1.00					
nationals of other EU countries	0.79	0.60–1.06	55.2%	ns ^(3)^	ns ^(3)^	ns ^(3)^
nationals of non-EU countries	1.20	0.80–1.81	87.4%	ns ^(3)^	ns ^(3)^	0.075
Smoking habits (smoker vs. non-smoker)						
nationals	1.00					
nationals of other EU countries	0.65	0.50–0.85	46.9%	-	-	-
nationals of non-EU countries	1.10	0.57–2.11	95.5%	ns ^(3)^	ns ^(3)^	ns ^(3)^
Physical activity (active vs. inactive)						
nationals	1.00					
nationals of other EU countries	1.10	0.81–1.50	49.8%	-	-	-
nationals of non-EU countries	0.95	0.56–1.62	92.5%	0.065	ns ^(3)^	ns ^(3)^
Attendance of breast cancer screening (yes vs. no)						
nationals	1.00					
nationals of other EU countries	0.66	0.31–1.39	60.6%	ns ^(3)^	ns ^(3)^	ns ^(3)^
nationals of non-EU countries	0.34	0.16–0.71	64.0%	ns ^(3)^	ns ^(3)^	ns ^(3)^
Attendance of cervical cancer screening (yes vs. no)						
nationals	1.00					
nationals of other EU countries	0.61	0.49–0.77	0.0%	-	-	-
nationals of non-EU countries	0.38	0.20–0.73	89.9%	ns ^(3)^	0.039	ns ^(3)^

^(1)^*p*-values were reported only when lower than 0.10. No meta-regression model was fitted when I^2^ < 50%. ^(2)^ HL, Health Literacy. ^(3)^ ns, not significant.

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
