# Peer review of "Health Literacy as a Shared Capacity: Does the Health Literacy of a Country Influence the Health Disparities among Immigrants?"

_ijerph, 2020, doi:10.3390/ijerph17041149_

Round 1

Reviewer 1 Report

a) This research has some misleadings: for instance the HLS-EU study in

- Austria does not cover the all country

- Greece does not cover the country but only Athens. Therefore sentences like 

160 the visual inspection of the forest plot suggested that Greece behaved as an outlier

are not contextualized and drift conclusions away from the basic statistical evidence

Therefore when addressing the limits of this study it should be deeply explored more than just

299 "limits of this study are partly related to the limited number of countries involved".

b) There is a lack of support for associations like

" health-literate populations have 285 the skills and resources necessary to address health concerns through civic engagement". 

What is civic engagement and where the evidence comes from for this sentence

c) the use of the word "race" should not be tolerated although it is a prevailing reality in research

"in persistent racial/ethnic health disparities".

Should only be used ""in persistent ethnic health disparities". 

Author Response

Thanks for the comments. Here the point-by-point reply:

This research has some misleadings: for instance the HLS-EU study in Austria does not cover the all country, and in Greece does not cover the country but only Athens.

Reply: thanks for the comment. In the manuscript, information about sampling procedure in the HLS-EU study has been added, as described in literature (Sørensen, K. et al. Health literacy in Europe: comparative results of the European health literacy survey (HLS-EU). Eur J Public Health 2015, 25(6), 1053-8). [lines 78-86]

Therefore sentences like : “160 the visual inspection of the forest plot suggested that Greece behaved as an outlier” are not contextualized and drift conclusions away from the basic statistical evidence.

Reply: We recognize that formal statistical tests exist to identify outliers, and that the procedure could have been made systematic from the beginning by applying a leave-one-out sensitivity analysis. However, given the limited number of “points” (n=5, i.e. one point for each country) in all meta-analysis models that were fitted, we believed that visually inspecting the forest plots would guarantee a good enough accuracy in identifying outliers. However, following this comment, we decided to run formal sensitivity analyses for all meta-analysis models, but results were, as expected, largely unchanged.

Therefore when addressing the limits of this study it should be deeply explored more than just: “299 "limits of this study are partly related to the limited number of countries involved".

Reply: In the limits, comments regarding the differences in sampling procedures in the HLS-EU survey have been added [lines 321-323]

There is a lack of support for associations like: “health-literate populations have 285 the skills and resources necessary to address health concerns through civic engagement". What is civic engagement and where the evidence comes from for this sentence

Reply: comments of civic engagement and on its relationship with health literacy have been added [lines 284-302] 

The use of the word "race" should not be tolerated although it is a prevailing reality in research. E.g. "in persistent racial/ethnic health disparities" should only be used ""in persistent ethnic health disparities". 

Reply: the sentence has been changed as suggested [line 342] 

Reviewer 2 Report

This is an interesting and clearly written paper about an important research topic for health literacy, distributed health literacy for immigrants. The set up and discussion are very good, but the analyses are complex and seem odd. I want to make sure I have it correct as there are not strong findings in the final models.

First, to show the hypothesized effect, you need to have differences for immigrants to EU citizens across the variety of health outcomes with differences across countries in this. Then you need to have HL at the country level explain this heterogeneity. In fact, many of the outcomes did not meet the first criteria. HL did not explain heterogeneity of any outcomes at the level of significance except for breast cancer.

Some other questions:

While I do agree the final models are unique, I am not sure about the basic information, for instance, Tables 2 and 3. I understand that you ran new analyses using publicly available information, but hasn't this basic descriptive information been published before for the cross-country comparisons from the European HL Survey somewhere? I would be clear about what is novel here and what is not and make sure permissions are in place for those that are not. If this is all novel, maybe a footnote explaining this would be helpful. Table 4 is the first place where the descriptive information seemed likely to be novel to this paper.

Some specifics:

The information in Table 1 is all individual level, yes? I would say so.

The only country-level variable was health literacy? This seems problematic. Countries with lower health literacy on average might, for instance, be poorer and have less social services that would be related to health care access for immigrants. I think you ideally need more country level variables in the model to control for other factors of variation.

Do you not have individual health literacy in model? The model has to at the very least have individual and contextual health literacy both. You do not think health literacy only works on the contextual level. You explain the interaction of individual and distributed h lit in the introduction.

There are some formatting errors. One of the page 2 of 21 still has this one line 239-241: "This section may be divided by subheadings. It should provide a concise and precise description of the experimental results, their interpretation as well as the experimental conclusions that can be drawn.

Please add more information about the limitations of the study and consider creating more meaningful models. Or consider just using this a think piece (take out the methods/models) as the intro and discussion seem much stronger to me than this analysis. 

Author Response

Thanks for the comments. Here the point-by-point reply:

This is an interesting and clearly written paper about an important research topic for health literacy, distributed health literacy for immigrants. The set up and discussion are very good, but the analyses are complex and seem odd. I want to make sure I have it correct as there are not strong findings in the final models.

        First, to show the hypothesized effect, you need to have differences for immigrants to EU citizens across the variety of health outcomes with differences across countries in this. Then you need to have HL at the country level explain this heterogeneity. In fact, many of the outcomes did not meet the first criteria. HL did not explain heterogeneity of any outcomes at the level of significance except for breast cancer.

Reply: Thank you for this comment. In fact, what this Reviewer describes in the paragraph above is exactly what we meant to do. Namely, for each of the outcomes that were studied, we merged country-specific OR (i.e. OR comparing immigrants vs. local citizens within each country) into a summary OR using meta-regression, calculated the I2 statistics as a measure of the variability of the association between countries, and then used meta-regression to assess whether the differences between country-specific ORs could be explained by (i.e. would correlate with) the HL measured at country level. It is true that the first criterion (“variety… across countries”) was not met for all outcomes, and in fact, a meta-regression model was only fitted when I2 was above 50%, which is considered (somewhat arbitrarily, we acknowledge, but widely in the literature) as a sign of substantial heterogeneity.

Please note that the whole procedure (country specific ORs, meta-analysis, estimation of the heterogeneity, and meta-regression to correlate with country-level HR) was separately performed for each of the study outcomes. Instead, no attempt was made to assess whether the association between migrant status and each study outcome was different for the different outcomes in study, as this was largely expected based on a priori considerations (i.e. we assumed that, and did not test whether, the association with migrant status was different for health status vs. BMI vs. smoking status etc.).       

Some other questions:

While I do agree the final models are unique, I am not sure about the basic information, for instance, Tables 2 and 3. I understand that you ran new analyses using publicly available information, but hasn't this basic descriptive information been published before for the cross-country comparisons from the European HL Survey somewhere? I would be clear about what is novel here and what is not and make sure permissions are in place for those that are not. If this is all novel, maybe a footnote explaining this would be helpful. Table 4 is the first place where the descriptive information seemed likely to be novel to this paper.

Reply: in our study, we obtained the microdata (i.e. the database) of the European Health Interview Survey (EHIS) of Eurostat. For this reason, we ran the statistics reported in Table 2. For what concern the HL, data was obtained consulting the published results (i.e. HLS-EU Consortium, 2012) and reported in Table 3 - with reference citation – in order to allow the readers to have immediately available the information we have used in the analysis. These aspects have been clarified in the main text [footnote at page 3] and in the table 2 caption.

Some specifics:

The information in Table 1 is all individual level, yes? I would say so.

Reply: yes, they are. Information has been added in the caption [line 105]

The only country-level variable was health literacy? This seems problematic. Countries with lower health literacy on average might, for instance, be poorer and have less social services that would be related to health care access for immigrants. I think you ideally need more country level variables in the model to control for other factors of variation.

Reply: this is a very interesting point, thank you for the comment. Our intention was to perform an explorative study aiming in assessing the impact of the HL of a country on health inequalities of immigrants. Other studies have already investigated the relationship between country-level determinants and immigrants’ integration, access to healthcare and/or health, highlighting significant associations (see, for instance: Malmusi, 2014; Nguyen, 2018; Giannoni, 2016). Moreover, our previous study published in  IJPH (Lorini, 2018) have found that some variables at the country level, such as the percentage of population with post-secondary education, the reading performance for 15-year-old students, the presence of a national breast or cervical cancer screening program, the unemployment rate, the Gross Domestic Product, the Gini coefficient, the rank of the Euro Patient Empowerment Index, and the expenditure on social protection were significantly associated with HL, also measured at the country-level. In that previous study, according to Sorensen framework on HL (Sorensen, 2012), all these variables were considered as antecedents of HL, namely factors that influence the HL at the population level. For these reasons we have decided not to include in our analysis other country-level variables. We know that the network of determinants of health inequalities for immigrants is complex and we know also that in this study we have used a reductive model, but the purpose of our research was also to test a new, different approach in analysing such a multifaceted issue. Moreover, using more country-level (i.e. ecological) variables (i.e. gross domestic product, human development index, and others) would have been difficult even technical wise, since one could expect that many of these are correlated with one another, which in turn, would imply a too high level of collinearity in a meta-regression model based on only five points (i.e. countries). We have added some comments on this issue in the “limits” section and in the conclusion, according also with the other reviewer’s comments [lines 330-335; 337-338]

Do you not have individual health literacy in model? The model has to at the very least have individual and contextual health literacy both. You do not think health literacy only works on the contextual level. You explain the interaction of individual and distributed h lit in the introduction.

Reply: unfortunately, we do not have individual HL since this aspect has not been investigated in the European Health Interview Survey of Eurostat and the data linkage between EHIS and HLS-EU is not possible. We have added some comments in the “limits” section [lines 320-321]

There are some formatting errors. One of the page 2 of 21 still has this one line 239-241: "This section may be divided by subheadings. It should provide a concise and precise description of the experimental results, their interpretation as well as the experimental conclusions that can be drawn.

Reply: we have corrected the formatting errors.

Please add more information about the limitations of the study and consider creating more meaningful models. Or consider just using this a think piece (take out the methods/models) as the intro and discussion seem much stronger to me than this analysis. 

Reply: according to the previous comments, we have added more information about the limitations of the study, as well as on the future perspectives for research to improve the model. Finally, we have stressed that this paper describe a new, different approach in analysing such a multifaceted issue (a think piece) [lines 330-335]

Round 2

Reviewer 2 Report

Thank you for your responses. These are helpful in contextualizing the strengths and weaknesses of the study. I still think that country level health literacy alone is conceptually and methodologically problematic, but you have better acknowledged the weakness. I really like your study's introduction and discussion and think you really need better data to test the models you are interested in with more countries, more contextual variables besides HL, and individual HL. This would be so much stronger.

Please consider revising your study goal and the first 2 lines of the discussion to be more balanced about your approach (and why such an analysis as this might be missing from the literature). Country level HL isn't the only explanatory variable that would explain health disparities for immigrants at the national level, but if it's the only variable in the model, it's all you are testing. I added "THE" here to where you have "an" to make the point clear. 

Here is what you say now at the start of the discussion (with my change): "This study aims to assess the role of HL as THE ecological variable, which means that HL is considered as THE macro-level aspect in predicting health disparities among immigrants at the national level. To the best of our knowledge, no previous studies using this study design and methodological approach have been published till date, so comparisons with similar researches are hindered."

Can you change the way this is mentioned as a study goal and in the beginning of the discussion to be more balanced, as in your response to reviewers: "Our intention was to perform an explorative study aiming in assessing the impact of the HL of a country on health inequalities of immigrants...." And then build your discussion from there.  

Or if you don't wish to do this, please better explain why having just one variable as the country contextual variable is not methodologically and conceptually problematic. I understand why, with only 5 countries, it is technically challenging to include more country-level variables b/c of collinearity, but that doesn't make the conceptual problem (that everything different about these countries that might impact immigrants' health is only represented by HL) disappear. The notes in your response to reviewers are helpful to make it clear that you see this issue more fully than your text currently acknowledges. 

A few minor comments:

Why is results bold in the abstract?

Thank you for giving more context to the Eurostats data, but you can probably reduce that lengthy Eurostats explanation a bit.  

Author Response

Thank you for your responses. These are helpful in contextualizing the strengths and weaknesses of the study. I still think that country level health literacy alone is conceptually and methodologically problematic, but you have better acknowledged the weakness. I really like your study's introduction and discussion and think you really need better data to test the models you are interested in with more countries, more contextual variables besides HL, and individual HL. This would be so much stronger.

Please consider revising your study goal and the first 2 lines of the discussion to be more balanced about your approach (and why such an analysis as this might be missing from the literature). Country level HL isn't the only explanatory variable that would explain health disparities for immigrants at the national level, but if it's the only variable in the model, it's all you are testing. I added "THE" here to where you have "an" to make the point clear. 

Here is what you say now at the start of the discussion (with my change): "This study aims to assess the role of HL as THE ecological variable, which means that HL is considered as THE macro-level aspect in predicting health disparities among immigrants at the national level. To the best of our knowledge, no previous studies using this study design and methodological approach have been published till date, so comparisons with similar researches are hindered."

Can you change the way this is mentioned as a study goal and in the beginning of the discussion to be more balanced, as in your response to reviewers: "Our intention was to perform an explorative study aiming in assessing the impact of the HL of a country on health inequalities of immigrants...." And then build your discussion from there.  

Or if you don't wish to do this, please better explain why having just one variable as the country contextual variable is not methodologically and conceptually problematic. I understand why, with only 5 countries, it is technically challenging to include more country-level variables b/c of collinearity, but that doesn't make the conceptual problem (that everything different about these countries that might impact immigrants' health is only represented by HL) disappear. The notes in your response to reviewers are helpful to make it clear that you see this issue more fully than your text currently acknowledges. 

Reply: Thank you for the comments. We totally agree with you regarding strengths and limitation of our study. We have changed the text according your suggestions, both in the aim of the study (lines 68-73) and in the discussion (lines 245-248).

A few minor comments:

Why is results bold in the abstract?

Reply: it was a mistake. We have removed the bold

Thank you for giving more context to the Eurostats data, but you can probably reduce that lengthy Eurostats explanation a bit.  

Reply: Information about Eurostat data have been reduced